# Rapid Development of Competitive Translation Engines for Access to Multilingual COVID-19 Information

**Andy Way \***, **Rejwanul Haque**, **Guodong Xie, Federico Gaspari, Maja Popović** and **Alberto Poncelas**

ADAPT Centre, School of Computing, D09 Y074 Dublin, Ireland; rejwanul.haque@adaptcentre.ie (R.H.); guodong.xie@adaptcentre.ie (G.X.); federico.gaspari@adaptcentre.ie (F.G.); maja.popovic@adaptcentre.ie (M.P.); alberto.poncelas@adaptcentre.ie (A.P.)
**\*** Correspondence: andy.way@adaptcentre.ie

**Abstract:** Every day, more people are becoming infected and dying from exposure to COVID-19. Some countries in Europe like Spain, France, the UK and Italy have suffered particularly badly from the virus. Others such as Germany appear to have coped extremely well. Both health professionals and the general public are keen to receive up-to-date information on the effects of the virus, as well as treatments that have proven to be effective. In cases where language is a barrier to access of pertinent information, machine translation (MT) may help people assimilate information published in different languages. Our MT systems trained on COVID-19 data are freely available for anyone to use to help translate information (such as promoting good practice for symptom identification, prevention, and treatment) published in German, French, Italian, Spanish into English, as well as the reverse direction.

**Keywords:** machine translation; COVID-19; crisis translation; neural MT; automatic evaluation; human evaluation; online MT; rapid response MT

---

## 1. Introduction and Motivation

The COVID-19 virus was first reported in China in late December 2019, and the World Health Organization declared its outbreak a public health emergency of international concern on 30 January 2020, and subsequently a pandemic on 11 March 2020 (https://www.who.int/emergencies/diseases/novel-coronavirus-2019/events-as-they-happen). Despite initial doubts as to whether it could be passed from human to human, very quickly the virus took hold and at the time of writing, official data from multiple reputable sources consistently indicate that more than 6.5 million individuals have been infected, and at least 380,000 people have died worldwide due to COVID-19. (Based on official data available as of 5th June 2020 from sources such as https://www.who.int/emergencies/diseases/novel-coronavirus-2019/situation-reports, https://www.ecdc.europa.eu/en/geographical-distribution-2019-ncov-cases and https://coronavirus.jhu.edu/).

Some countries responded pretty quickly to the onset of COVID-19, imposing strict barriers on human movement in order to "flatten the curve" and avoid as much transmission of the virus as possible. Regrettably, others did not take advantage from the lessons learned initially in the Far East, with significant delays in implementing social distancing with concomitant rises in infection and death, particularly among the elderly.

Being an airborne disease, different countries were exposed to the virus at different times; some countries are starting to relax social distancing constraints and are allowing certain sections of the population back to work and school. Nonetheless, the virus remains rampant in many countries,

yet there is still time to absorb the lessons learned in other regions to try to keep infection and associated deaths at the lower end of projections.

However, much salient information online appears in a range of languages, so that access to information is restricted by people's language competencies. It has long been our view that in the 21st century, language cannot be a barrier to access of information. Accordingly, we decided to build a range of MT systems to facilitate access to multilingual content related to COVID-19. Given that Spain, France and Italy suffered particularly badly in Europe, it was important to include Spanish, French and Italian in our plans, so that lessons learned in those countries could be rolled out in other jurisdictions. The UK and the US have also suffered particularly badly, so English obviously had to be included. In contrast, Germany appears to have coped particularly well, so we wanted information written in German to be more widely available.

Accordingly, this document describes our efforts to build 8 MT systems—FIGS (French, Italian, German, Spanish. FIGS is a well-known term encompassing these languages in the localisation field) to/from English—using cutting-edge techniques aimed specifically at making available health and related information (e.g., promoting good practice for symptom identification, prevention, treatment, recommendations from health authorities, etc.) concerning the COVID-19 virus both to medical experts and the general public. In the current situation, the volume of information to be translated is huge and is relevant (hopefully) only for a relatively short span of time; relying wholly on human professional translation is not an option, both in the interest of timeliness and due to the enormous costs that would be involved. By making the engines publicly available, we are empowering individuals to access multilingual information that otherwise might be denied them; by ensuring that the MT quality is comparable to that of well-known online systems, we are allowing users to avail of high-quality MT with none of the usual privacy concerns associated with using online MT platforms. Furthermore, we note interesting strengths and weaknesses of our engines following a detailed comparison with online systems. Finally, of course, by building our own engines, we retain control over the systems, which facilitates continuous incremental improvement of the models via feedback and by the availability of additional training data, or novel scientific breakthroughs in the field of Neural MT (NMT).

The remainder of the paper explains what data we sourced to train these engines, how we trained and tested them to ensure good performance, and our efforts to make them available to the general public. It is our hope that these engines will be helpful in the global fight against this pandemic, so that fewer people are exposed to this disease and its deadly effects.

## 2. Ethical Considerations in Crisis-Response Situations

There are of course a number of challenges and potential dangers involved in rapid development of MT systems for use by naive and vulnerable users to access potentially sensitive and complex medical information.

There have been alarmingly few attempts to provide automatic translation services for use in crisis scenarios. The best-known example is Microsoft's rapid response to the 2010 Haiti earthquake [1], which in turn led to a cookbook for MT in crisis scenarios [2].

In that paper, Lewis et al. [2], p. 501 begin by stating that "*MT is an important technology in crisis events, something that can and should be an integral part of a rapid-response infrastructure . . . If done right, MT can dramatically increase the speed by which relief can be provided*". They go on to say the following:

> "*We strongly believe that MT is an important technology to facilitate communication in crisis situations, crucially since it can make content in a language spoken or written by a local population accessible to those that do not know the language*" [p. 501]

They also note [pp. 503–504] that "While translation is not [a] widely discussed aspect of crisis response, it is 'a perennial hidden issue'" [3]:

> *"Go and look at any evaluation from the last ten or fifteen years. 'Recommendation: make effective information available to the government and the population in their own language.' We didn't do it . . . It is a consistent thing across emergencies."* Brendan McDonald, UN OCHA in [3].

While it is of course regrettable that natural disasters continue to occur, these days we are somewhat better prepared to respond when humanitarian crises such as COVID-19 occur, thanks to work on translation in crisis situations such as INTERACT. (https://sites.google.com/view/crisistranslation/home) Indeed, Federici et al. [4] issue a number of recommendations within that project which we have tried to follow in this work. While they apply mainly to human translation provision in crisis scenarios, they can easily be adapted to the use of MT, as in this paper.

The main relevant recommendation ([Recommendation 1] in their report; we use their numbering in what follows) is that "Emergency management communication policies should include provision for translation" [p. 2], which we take as an endorsement of our approach here. The provision of MT systems has the potential to help:

- "improve response, recovery and risk mitigation [by including] mechanisms to provide accurate translation" [Recommendation 1a, p. 8]
- "address the needs of those with heightened vulnerabilities [such as] . . . the elderly" [Recommendation 1b, p. 9]
- those "responsible for actioning, revising and training to implement . . . translation policy within . . . organization[s]" [Recommendation 2a, p. 9]

Federici et al. [4] note that "the right to translated information in managing crises must be a part of 'living policy and planning documents' that guide public agency actions" [Recommendation 2b, p. 9], and that people have a "right to translated information across all phases of crisis and disaster management" [Recommendation 4a, p. 9]. We do not believe that either of these have been afforded to the public at large during the current crisis, but our provision of MT engines has the potential to facilitate both of these requirements.

[Recommendation 7a, p. 10] notes that "Translating in one direction is insufficient. Two-way translated communication is essential for meeting the needs of crisis and disaster-affected communities." By allowing translation in both directions (FIGS-to-English as well as English-to-FIGS), we are facilitating two-way communication, which would of course be essential in a patient-carer situation, for example.

By making translation available via MT rather than via human experts, we are helping avoid the need for "training for translators and interpreters . . . includ[ing] aspects of how to deal with traumatic situations" [Recommendation 8d, p. 11], as well as translators being exposed to traumatic situations altogether.

Finally, as we describe below, we have taken all possible steps to ensure that the quality of our MT systems is as good as it can be at the time of writing. Using leading online MT systems as baselines, we demonstrate comparable performance—and in some cases improvements over some of the online systems—and document a number of strengths and weaknesses of our models as well as the online engines. We deliberately decided to release our systems as soon as good performance was indicated both by automatic and human measures of translation quality; aiming for fully optimized performance would have defeated the purpose of making the developed MT systems publicly available as soon as possible to try and mitigate the adverse effects of the ongoing international COVID-19 crisis.

## 3. Assembling Relevant Data Sets

NMT [5] is acknowledged nowadays as the leading paradigm in MT system-building. However, compared to its predecessor (Statistical MT (SMT): Koehn et al. [6]), NMT requires even larger amounts of suitable training data in order to ensure good translation quality. For example, Koehn and Knowles [7] show that using BLEU [8] as the evaluation metric, English-to-Spanish NMT systems start to

outperform SMT with around 15 million words of parallel data, and can beat an SMT system with a huge two billion word in-domain language model when trained on 385 million words of parallel data.

It is well-known in the field that optimal performance is most likely to be achieved by sourcing large amounts of training data which are as closely aligned with the intended domain of application as possible [9,10]. Accordingly, we set out to assemble large amounts of good quality data in the language pairs of focus dedicated to COVID-19 and the wider health area.

Table 1 shows how much data was gathered for each of the language-pairs. We found this in a number of places, including: (An additional source of data is the NOW Corpus (https://www.english-corpora.org/now/), but we did not use this as it is not available for free).

- TAUS Corona Crisis Corpus (https://md.taus.net/corona)
- EMEA Corpus (http://opus.nlpl.eu/EMEA.php)
- Sketch Engine Corpus (https://www.sketchengine.eu/covid19/)
- ParaCrawl Corpus (http://opus.nlpl.eu/ParaCrawl-v5.php)
- Wikipedia Corpus (http://opus.nlpl.eu/Wikipedia-v1.0.php)

**Table 1.** Training data sizes for each language pair.

| Language Pair | Parallel Sentences |
| --- | --- |
| DE–EN | 29,433,082 |
| ES–EN | 10,689,702 |
| FR–EN | 11,298,918 |
| IT–EN | 10,241,827 |

**The TAUS Corona data** comprises a total of 885,606 parallel segments for EN–FR, 613,318 sentence-pairs for EN–DE, 381,710 for EN–IT, and 879,926 for EN–ES. However, as is good practice in the field, we apply a number of standard cleaning routines to remove 'noisy' data, e.g., removing sentence-pairs that are too short, too long or which violate certain sentence-length ratios. This results in 343,854 sentence-pairs for EN–IT (i.e., 37,856 sentence-pairs—amounting to 10% of the original data—are removed), 698,857 for EN–FR (186,719 sentence-pairs, 21% of the original), 791,027 for EN–ES (88,899 sentence-pairs, 10%), and 551327 for EN–DE (61,991 sentence-pairs, 10%).

**The EMEA Corpus** (a parallel corpus comprised of documents from the European Medicines Agency that focuses on the wider health area) comprises 499,694 segment-pairs for EN–IT, 471,904 segment-pairs for EN–ES, 454,730 segment-pairs for EN–FR, and 1,108,752 segment-pairs for EN–DE.

**The Sketch Engine Corpus** comprises 4,736,815 English sentences.

**The ParaCrawl Corpus** (parallel sentences crawled from the Web) comprises 19,675,137 segment-pairs for EN–IT, 34,561,031 segment-pairs for EN–ES, 45,941,170 segment-pairs for EN–FR, and 58,815,994 segment-pairs for EN–DE.

**The Wikipedia Corpus** (parallel sentences extracted from Wikipedia) comprises 957,004 segment-pairs for EN–IT, 1,811,428 segment-pairs for EN–ES, 818,302 segment-pairs for EN–FR, and 2,459,662 segment-pairs for EN–DE.

Note that we used a part of the ParaCrawl and Wikipedia corpora following the state-of-the-art sentence selection strategy of Axelrod et al. [11], which is detailed in Section 4.3.

There have been suggestions (e.g., on Twitter) that the data found in some of the above-mentioned corpora, especially those that were recently released to support rapid development initiatives such as the one reported in this paper, is of variable and inconsistent quality, especially for some language pairs. For example, a quick inspection of samples of the TAUS Corona Crisis Corpus for English–Italian revealed that there are identical sentence pairs repeated several times, and numerous segments that do not seem to be directly or explicitly related to the COVID-19 infection *per se*, but appear to be about medical or health-related topics in a very broad sense; in addition, occasional irrelevant sentences

about computers being infected by software viruses come up, that may point to unsupervised text collection from the Web.

Nonetheless, we are of course extremely grateful for the release of the valuable parallel corpora that we were able to use effectively for the rapid development of the domain-specific MT systems that we make available to the public as part of this work; we note these observations here as they may be relevant to other researchers and developers who may be planning to make use of such resources for other corpus-based applications.

## 4. Experiments and Results

This section describes how we built a range of NMT systems using the data described in the previous section. We evaluate the quality of the systems using BLEU and chrF [12]. Both are match-based metrics (so the higher the score, the better the quality of the system) where the MT output is compared against a human reference. The way this is typically done is to 'hold out' a small part of the training data as test material, (Including test data in the training material will unduly influence the quality of the MT system, so it is essential that the test set is disjoint from the data used to train the MT engines) where the human-translated target sentence is used as the reference against which the MT hypothesis is compared. BLEU does this by computing $n$-gram overlap, i.e., how many words, 2-, 3- and 4-word sequences are contained in both the reference and hypothesis. (Longer $n$-grams carry more weight, and a penalty is applied if the MT system outputs translations which are 'too short'). There is some evidence [13] that word-based metrics such as BLEU are unable to sufficiently demonstrate the difference in performance between MT systems, (For some of the disadvantages of using string-based metrics, we refer the reader to Way [14]). So in addition, we use chrF, a character-based metric which is more discriminative. Instead of matching word $n$-grams, it matches character $n$-grams (up to 6). Since Popović [15] shows that the best option is to give twice as much weight to recall, we used this variant (chrF2) in this work.

### 4.1. Building Baseline MT Systems

In order to build our NMT systems, we used the MarianNMT [16] toolkit. The NMT systems are Transformer models [17]. In our experiments we followed the recommended best set-up from Vaswani et al. [17]. The tokens of the training, evaluation and validation sets are segmented into sub-word units using the byte-pair encoding technique of Sennrich et al. [18]. We performed 32,000 join operations.

Our training set-up is as follows. We consider the size of the encoder and decoder layers to be 6. As in Vaswani et al. [17], we employ residual connection around layers [19], followed by layer normalisation [20]. The weight matrix between embedding layers is shared, similar to Press and Wolf [21]. Dropout [22] between layers is set to 0.1. We use mini-batches of size 64 for update. The models are trained with the Adam optimizer [23], with the learning-rate set to 0.0003 and reshuffling the training corpora for each epoch. As in Vaswani et al. [17], we also use the learning rate warm-up strategy for Adam. The validation on the development set is performed using three cost functions: cross-entropy, perplexity and BLEU. The early stopping criterion is based on cross-entropy; however, the final NMT system is selected as per the highest BLEU score on the validation set. The beam size for search is set to 12.

The performance of our engines is described in Table 2. Testing on a set of 1000 held-out sentence pairs, we obtain a BLEU score of 50.28 for IT-to-EN. For the other language pairs, we also see good performance, with all engines bar EN-to-DE—a notoriously difficult language pair—obtaining BLEU scores in the range of 44–50, already indicating that these rapidly-developed MT engines could be deployed 'as is' with some benefit to the public in several countries affected by the COVID-19 epidemic.

**Table 2.** Performance of the baseline NMT systems.

|  | BLEU | | chrF | |
| --- | --- | --- | --- | --- |
|  | **TAUS** | **Reco** | **TAUS** | **Reco** |
| Italian-to-English | 50.28 | 51.02 | 71.47 | 72.25 |
| German-to-English | 44.05 | 38.52 | 60.70 | 57.81 |
| Spanish-to-English | 50.89 | 31.42 | 71.92 | 54.84 |
| French-to-English | 46.17 | 35.78 | 69.20 | 54.78 |
| English-to-Italian | 45.21 | 47.30 | 69.32 | 71.67 |
| English-to-German | 35.58 | 37.15 | 57.63 | 55.47 |
| English-to-Spanish | 46.72 | 32.07 | 68.80 | 57.35 |
| English-to-French | 43.21 | 34.26 | 67.55 | 61.59 |

We also created specific test sets on COVID-19 recommendations (cf. Table 3). These test sets range from 73–104 sentences. The German test set was taken from the website of The Robert Koch Institute, the government's central scientific institution in the field of biomedicine. (https://www.rki.de/EN/Home/homepage_node.html). The French test set was taken from the website of Santé publique France (Public Health in France). (https://www.santepubliquefrance.fr/). Both texts were originally written in German and French, respectively, but English translations were available, too. The Spanish test set consists of a collection of sentences from recommendations on the website of the Ministry of Health of Spain (https://www.mscbs.gob.es/profesionales/saludPublica/ccayes/alertasActual/nCov-China/documentos/20.03.20_AislamientoDomiciliario_COVID19.pdf and https://www.mscbs.gob.es/profesionales/saludPublica/ccayes/alertasActual/nCov-China/documentos/20200319_Decalogo_como_actuar_COVID19.pdf) and the World Health Organization. (https://www.who.int/es/emergencies/diseases/novel-coronavirus-2019/advice-for-public). The Italian test set was extracted from a regularly updated webpage with Frequently Asked Questions and recommended best practices on COVID-19 on the website of the Italian Ministry of Health, (http://www.salute.gov.it/portale/nuovocoronavirus/dettaglioFaqNuovoCoronavirus.jsp?id=228) that had its own official English translation. (http://www.salute.gov.it/portale/nuovocoronavirus/dettaglioFaqNuovoCoronavirus.jsp?lingua=english&id=230).

On these separate 'Reco' test sets, MT performance is still reasonable. For IT-to-EN and EN-to-IT, the performance even increases a little according to both BLEU and chrF, but in general translation quality drops off by 10 BLEU points or more. Notwithstanding the fact that these test sets are much shorter than the TAUS sets, we reiterate that the primary motivation behind this work was to build as quickly as we could a range of good-quality MT systems capable of translating information related to the COVID-19 virus. (One may wonder why we eschewed the 'massively multilingual' approach of Arivazhagan et al. [24]. Firstly, note that even with what might be deemed to be 'moderate' amounts of data, the hardware requirements are enormous. Secondly, there is evidence [25,26] to suggest that multilingual models are not guaranteed to beat individual MT systems for specific language pairs. Nonetheless, it remains an open question as to whether transfer learning techniques could have been used to fine-tune on the different language pairs, as in Neubig and Hu [27]; we leave this for future work). Accordingly, the systems' performance on the Reco sets is more indicative of their ability on related but more out-of-domain data; being extracted from the TAUS data set itself, one would expect better performance on truly in-domain data, but this is a less important factor.

**Table 3.** Statistics of the "recommendations" test sets.

|  | **Sent** | **Words (FIGS)** | **Words (EN)** |
|---|---|---|---|
| Italian–English | 100 | 2915 | 2728 |
| German–English | 104 | 1255 | 1345 |
| Spanish–English | 73 | 1366 | 1340 |
| French–English | 81 | 1013 | 965 |

Note that as expected, scores for translation into English are consistently higher compared to translation from English. This is because English is relatively morphologically poorer than the other four languages, and it is widely recognised that translating into morphologically-rich languages is a (more) difficult task.

*4.2. How Do Our Engines Stack Up against Leading Online MT Systems?*

In order to examine the competitiveness of our engines against leading online systems, we also translated the Reco test sets using Google Translate, (https://translate.google.com/). Bing Translator, (https://www.bing.com/translator) and Amazon Translate. (https://aws.amazon.com/translate/. (i) Note that while Google Translate and Bing Translator are free to use, Amazon Translate is free to use for a period of 12 months, as long as you register online; (ii) One of the reviewers suggested that we also look at DeepL: https://www.deepl.com/en/translator. The free online version available at this URL is almost unusable for even quite small amounts of translation, as here. Nonetheless, its results are extremely impressive. The BLEU scores on the 'Reco' test sets are as follows: DE-to-EN: 38.3; IT-to-EN: 68.9; FR-to-EN: 42.9; and ES-to-EN: 38.6. For all bar the latter language pair, these results are the best, and in some cases by some distance. Given the turnaround requirements of the journal, we were unable to perform a human evaluation in such a short time, but the actual output clearly deserves to be looked at; (iii) It should also be noted that Systran have in the interim published a series of systems dedicated to COVID-19 material: https://www.systransoft.com/systran/news-and-events/specialized-corona-crisis-corpus-models/, but again, we have been unable to test their performance in the limited time available). The results for the 'into-English' use-cases appear in Table 4.

**Table 4.** Performance of online NMT systems.

|  |  | **Google** | **Amazon** | **Bing** |
|---|---|---|---|---|
| DE-to-EN | BLEU | 35.1 | 33.9 | 35.8 |
|  | chrF | 57.7 | 56.2 | 57.3 |
| IT-to-EN | BLEU | 61.2 | 58.9 | 59.2 |
|  | chrF | 79.1 | 76.8 | 76.9 |
| FR-to-EN | BLEU | 40.4 | 36.3 | 38.4 |
|  | chrF | 58.2 | 54.4 | 56.9 |
| ES-to-EN | BLEU | 38.0 | 38.9 | 37.0 |
|  | chrF | 58.9 | 59.5 | 57.9 |

As can be seen, for DE-to-EN, in terms of BLEU score, Bing is better than Google, and 1.9 points (5.6% relatively) better than Amazon. In terms of chrF, Amazon is still clearly in 3rd place, but this time Google's translations are better than those of Bing.

For both IT-to-EN and FR-to-EN, Google outperforms both Bing and Amazon according to both BLEU and chrF, with Bing in 2nd place.

However, for ES-to-EN, Amazon's system obtains the best performance according to both BLEU and chrF, followed by Google and then Bing.

If we compare the performance of our engines against these online systems, in general, for all four language pairs, our performance is worse; for DE-to-EN, the quality of our MT system in terms of BLEU is better than Amazon's, but for the other three language pairs we are anything from 0.5 to 10 BLEU points worse.

This demonstrates clearly how strong the online baseline engines are on this test set. However, in the next section, we run a set of experiments which show improved performance of our systems, in some cases comparable to those of these online engines.

### 4.3. Using "Pseudo In-domain" Parallel Sentences

The previous sections demonstrate clearly that with some exceptions, on the whole, our baseline engines significantly underperform compared to the major online systems.

However, in an attempt to improve the quality of our engines, we extracted parallel sentences from large bitexts that are similar to the styles of texts we aim to translate, and use them to fine-tune our baseline MT systems. For this, we followed the state-of-the-art sentence selection approach of Axelrod et al. [11] that extracts 'pseudo in-domain' sentences from large corpora using bilingual cross-entropy difference over each side of the corpus (source and target). The bilingual cross-entropy difference is computed by querying in- and out-of-domain (source and target) language models. Since our objective is to facilitate translation services for recommendations for the general public, non-experts, and medical practitioners in relation to COVID-19, we wanted our in-domain language models to be built on such data. Accordingly, we crawled sentences from a variety of government websites (e.g., HSE, (https://www2.hse.ie/conditions/coronavirus/coronavirus.html) NHS, (https://www.nhs.uk/conditions/coronavirus-covid-19/). Italy's Ministry of Health and National Institute of Health, (https://www.salute.gov.it/nuovocoronavirus and https://www.epicentro.iss.it/coronavirus/, respectively). Ministry of Health of Spain, (https://www.mscbs.gob.es) and the French National Public Health Agency) that offer recommendations and information on COVID-19. In Table 5, we report the statistics of the crawled corpora used for in-domain language model training.

**Table 5.** Statistics of the crawled corpora used for in-domain language model training.

|         | Words   |
| ------- | ------- |
| English | 24,118  |
| French  | 779,654 |
| Italian | 231,157 |
| Spanish | 193,797 |

The so-called 'pseudo in-domain' parallel sentences that were extracted from the closely related out-of-domain data (such as the EMEA Corpus) were appended to the training data. Finally, we fine-tuned our MT systems on the augmented training data.

In addition to the EMEA Corpus, we made use of ParaCrawl and Wikipedia data from OPUS (http://opus.nlpl.eu/) [28] (cf. Section 3) which we anticipate containing sentences related to general recommendations similar to the styles of the texts we want to translate. We merged the ParaCrawl and Wikipedia corpora, which from now on we refer to as the "ParaWiki Corpus". First, we chose the Italian-to-English translation direction to test our data selection strategy on EMEA and ParaWiki corpora. Once the best set-up had been identified, we would use the same approach to prepare improved MT systems for the other translation pairs.

We report the BLEU and chrF scores obtained on a range of different Italian-to-English NMT systems on both the TAUS and Reco test sets in Table 6.

**Table 6.** The Italian-to-English NMT systems. SEC: Sketch Engine Corpus.

|   |           | BLEU | | chrF | |
|---|-----------|------|------|------|------|
|   |           | **TAUS** | **Reco** | **TAUS** | **Reco** |
| 1 | Baseline     | 50.28 | 51.02 | 71.47 | 72.25 |
| 2 | 1 + EMEA     | 50.43 | 52.72 | 71.47 | 73.25 |
| 3 | 1 + SEC      | 47.44 | 51.67 | 69.66 | 72.51 |
| 4 | 2 + ParaWiki | 50.96 | 54.97 | 71.85 | 75.35 |
| 5 | 3 + 4        | 50.56 | 57.77 | 71.78 | 76.29 |
| 6 | 3 + 4*       | 49.54 | 57.10 | 71.13 | 76.30 |
| 7 | Ensemble     | 50.60 | 58.16 | 71.74 | 76.91 |

The second row in Table 6 represents the Italian-to-English baseline NMT system fine-tuned on the training data from the TAUS Corona Crisis Corpus along with 300 K sentence-pairs from the EMEA Corpus using the aforementioned sentence-selection strategy. We expect that training for some extra iterations on a well-considered sample of in-domain and generic data should both improve performance, and avoid overfitting on the TAUS+EMEA data. While we see that this strategy does lead to some improvements on both test sets over the baseline, the gains are only moderate, and are not statistically significant using bootstrap resampling with a 99% confidence level ($p < 0.01$) [29]. It is an open question whether including the full EMEA set would improve performance, but adding in the 300 K most similar sentences barely helps; even if we assume that 700 K sentence-pairs are disregarded by our approach, after cleaning and other filtering operations, the remaining subset would be far smaller than 700 K. Nonetheless, we leave this experiment for future work, in order to check definitively whether the additional data proves to be useful.

Currey et al. [30] generated synthetic parallel sentences by copying target sentences to the source. This method was found to be useful where scripts are identical across the source and target languages. Note too that they showed back-translation to be less effective in low-resource settings where it is hard to train a good back-translation model, so we propose to use the method of Currey et al. [30] instead.

In our case, since the Sketch Engine Corpus is prepared from the COVID-19 Open Research Dataset (CORD-19), (A free resource of over 45,000 scholarly articles, including over 33,000 with full text, about COVID-19 and the coronavirus family of viruses, cf. https://pages.semanticscholar.org/coronavirus-research) it includes keywords and terms related to COVID-19, which are often used verbatim across the languages of study in this paper, so we contend that this strategy can help translate terms correctly. Accordingly, we carried out an experiment by adding sentences of the Sketch Engine Corpus (English monolingual corpus) to the TAUS Corona Crisis Corpus following the method of Currey et al. [30], and fine-tune the model on this training set. The scores for the resulting NMT system are shown in the third row of Table 6. While this method also brings about moderate improvements on the Reco test set, it is not statistically significant. Interestingly, this approach also significantly lowers the system's performance on the TAUS test set, according to both metrics. Nonetheless, given the urgency of the situation in which we found ourselves, where the MT systems needed to be built as quickly as possible, the approach of Currey et al. [30] can be viewed as a worthwhile alternative to the back-translation strategy of Sennrich et al. [31] which needs a high-quality back-translation model to be built, and the target corpus to be translated into te source language, both of which are time-demanding tasks. (Note that albeit on a different language pair (English-to-German), Khayrallah and Koehn [32] observe that untranslated sentences (i.e., copying target sentences to the source) can severely negatively impact NMT systems in mid- to high-resource scenarios. However, they note that "short segments, untranslated source sentences and wrong source language have little impact on either [NMT or SMT]" [p. 78]. In our case, we tested the approach of Currey et al. [30] only when the source side is the copy of the target, and like them, found it to be effective).

In our next experiment, we took five million low-scoring (i.e., similar to the in-domain corpora) sentence-pairs from ParaWiki, added them to the training data, and fine-tuned the baseline model on it. As can be seen from row 4 in Table 6 (i.e., 2 + ParaWiki), the use of ParaWiki sentences for fine-tuning has a positive impact on the system's performance, as we obtain a statistically significant 3.95 BLEU point absolute gain (corresponding to a 7.74% relative improvement) on the Reco test set over the baseline. This is corroborated by the chrF score.

We then built a training set from all data sources (EMEA, Sketch Engine, and ParaWiki Corpora), and fine-tuned the baseline model on the combined training data. This brings about a further statistically significant gain on the Reco test set (6.75 BLEU points absolute, corresponding to a 13.2% relative improvement (cf. fifth row of Table 6, "3 + 4").

Our next experiment involved adding a further three million sentences from ParaWiki. This time, the model trained on augmented training data performs similarly to the model without the additional data (cf. sixth row of Table 6, "3 + 4*"), which is disappointing.

Our final model is built with ensembles of all eight models that are sampled from the training run (cf. fifth row of Table 6, "3 + 4"), and one of them is selected based on the highest BLEU score on the validation set. This brings about a further slight improvement in terms of BLEU on the Reco test set. Altogether, the improvement of our best model ('Ensemble' in Table 6, row 7) over our Baseline model is 7.14 BLEU points absolute, a 14% relative improvement. More importantly, while we cannot beat the online systems in terms of BLEU score, we are now in the same ballpark. More encouragingly still, in terms of chrF, our score is higher than both Amazon and Bing, although still a little way off compared to Google Translate.

Given these encouraging findings, we used the same set-up to build improved engines for the other translation pairs. The results in Table 7 show improvements over the Baseline engines in Table 2 for almost all language pairs on the Reco test sets: for DE-to-EN, the score dips a little, but for FR-to-EN, we see an improvement of 1.42 BLEU points (4% relative improvement), and for ES-to-EN by 2.09 BLEU points (6.7% relative improvement). While we still largely underperform compared to the scores for the online MT engines in Table 4, we are now not too far behind; indeed, for FR-to-EN, our performance is now actually better than Amazon's, by 0.9 BLEU (2.5% relative improvement) and 1.29 chrF points (2.4% relative improvement). For the reverse direction, EN-to-DE also drops a little, but EN-to-IT improves by 1.79 BLEU points (3.8% relatively better), EN-to-ES by 1.02 BLEU points (3.2% relatively better), and EN-to-FR by 2.22 BLEU points (6.5% relatively better). These findings are corroborated by the chrF scores.

**Table 7.** Our improved NMT systems.

| | BLEU | | chrF | |
|---|---|---|---|---|
| | **TAUS** | **Reco** | **TAUS** | **Reco** |
| Italian-to-English | 50.60 | 58.16 | 71.74 | 76.91 |
| German-to-English | 61.49 | 37.81 | 75.92 | 55.56 |
| Spanish-to-English | 49.99 | 33.51 | 71.55 | 56.36 |
| French-to-English | 46.41 | 37.20 | 69.66 | 55.67 |
| English-to-Italian | 45.89 | 49.09 | 69.95 | 72.61 |
| English-to-German | 54.41 | 36.77 | 73.17 | 54.99 |
| English-to-Spanish | 47.86 | 33.09 | 69.82 | 58.53 |
| English-to-French | 44.56 | 36.48 | 68.26 | 59.52 |

### 4.4. Human Evaluation

Using automatic evaluation metrics such as BLEU and chrF allows MT system developers to rapidly test the performance of their engines, experimenting with the data sets available and fine-tuning

the parameters to achieve the best-performing set-up. However, it is well-known [14] that where possible, human evaluation ought to be conducted in order to verify that the level of quality globally indicated by the automatic metrics is broadly reliable.

Accordingly, for all four 'into-English' language pairs, we performed a human evaluation on 100 sentences from the test set, comparing our system (indicated by "DCU" in the tables) against Google Translate and Bing Translator (cf. Table 8); we also inspected Amazon's output, but its quality was generally slightly lower, so in the interest of clarity it was not included in the comparisons discussed here. At the time of writing, the countries where the rate of infection of COVID-19 were the highest were the UK and US, so we concentrated more on translation into English rather than from English. As we initially tried to improve translation performance on Italian to English before rolling out the set-up to the other language directions (cf. Section 4.3), there is somewhat more analysis for this language pair than for the others.

In what follows, we illustrate the strengths and weaknesses of each system using examples from 100 sentences from each test set, according to a range of translational phenomena (such as lexical choice, fluency, and translation omission).

**Table 8.** Percentage of sentences translated by our systems which are adjudged to be better, worse or of the same quality compared to Google Translate and Bing Translator.

|  | Better Than Both | Same | Worse Than Either | Not Fully Clear |
|---|---|---|---|---|
| DE-EN | 19% | 50% | 15% | 16% |
| IT-EN | 5% | 54% | 26% | 15% |
| FR-EN | 5% | 50% | 18% | 27% |
| ES-EN | 4% | 68% | 17% | 11% |

### 4.4.1. German to English

Translation examples for German-to-English appear in Table 9. Despite the discrepancies in terms of automatic evaluation scores, our system often results in better lexical choice than the online ones (examples 1, 2 and 3). All three German words in question are ambiguous, and can have different English translations in different contexts. The German word in example 1, "übermitteln", could be translated as "transmit" (Google's choice) or "submit" (Bing's choice), but not in the given context of the cases which were actually "reported" (DCU's choice, and correct in this context). Example 2 shows another ambiguous German word, "unübersehbar", which can mean "obvious" but also "incalculable" or "inestimable". Google and Bing systems choose the word "unmistakable" which would be a good choice for the first sense ("obvious"), but in the given context, the second sense is correct (a vast incalculable number of regions). Similarly, the German word "schwer" in example 3 can generally be translated as "difficult" or "severe", but in the given context only the second sense is correct.

In many cases, our system outperforms one of the online systems, but not another (example 4). The German word "verbreitet" can mean "spread", "distributed", and also "popular". In the given context, Google's lexical choice is the best ("widespread"), DCU's is slightly worse ("distributed"), whereas Bing's is completely wrong ("used", in the sense of "popular").

The main advantage of the online systems is fluency, as shown in example 5, where our system treated the German verb "werden" as a future tense auxiliary verb rather than a passive voice auxiliary. Example 6 represents a very rare case where our engine omitted one part of the source sentence altogether, thus not conveying all of the original meaning in the translation.

**Table 9.** Translation examples for German-to-English.

| System | Example |
| --- | --- |
| (1) source | Unter den **übermittelten** COVID-19-Fällen |
| (1) reference | Of **notified** cases with a COVID-19 infection |
| google | Among the transmitted COVID-19 cases |
| bing | Among the COVID-19 cases submitted |
| dcu (best) | Among the reported COVID-19 cases |
| (2) source | in einer **unübersehbaren** Anzahl von Regionen weltweit. |
| (2) reference (shifted) | in many other, **not always well-defined** regions |
| google | in an unmistakable number of regions worldwide. |
| bing | in an unmistakable number of regions worldwide. |
| dcu (best) | in an innumerable number of regions worldwide. |
| (3) source | Bei einem Teil der Fälle sind die Krankheitsverläufe **schwer**, auch tödliche Krankheitsverläufe kommen vor. |
| (3) reference (shifted) | **Severe** and fatal courses occur in some cases |
| google | In some of the cases, the course of the disease is difficult, and fatal course of the disease also occurs. |
| bing | In some cases, the disease progressions are difficult, and fatal disease histories also occur. |
| dcu (best) | In some cases, the course of the disease is severe, including fatal cases. |
| (4) source | COVID-19 ist inzwischen weltweit **verbreitet**. |
| (4) reference (shifted) | Due to pandemic **spread**, there is a global risk of acquiring COVID-19. |
| google (best) | COVID-19 is now widespread worldwide. |
| bing (worse) | COVID-19 is now widely used worldwide. |
| dcu | COVID-19 is now widely distributed worldwide. |
| (5) source | Änderungen **werden** im Text in Blau **dargestellt** |
| (5) reference | Changes **are marked** blue in the text |
| google | Changes are shown in blue in the text |
| bing | Changes are shown in blue in the text |
| dcu (worst) | Changes will appear in blue text |
| (6) source | Bei 46.095 Fällen ist der **Erkrankungsbeginn nicht bekannt bzw. diese Fälle sind nicht symptomatisch erkrankt** |
| (6) reference (missing part) | In 46,095 cases, onset of symptoms is unknown *{MISSING}* |
| google | In 46,095 cases, the onset of the disease is unknown or these cases are not symptomatically ill |
| bing (best) | In 46,095 cases, the onset of the disease is not known or these cases are not symptomatic |
| dcu (worst) | In 46,095 cases, the onset or absence of symptomatic disease is not known {MISSING} |

### 4.4.2. French to English

Table 10 shows translation examples for French-to-English. Again, lexical choice is often better in our MT hypotheses (example 1). "Regrouper" is a false friend which does not mean "regrouping", as translated by Google and Bing, but "grouping" or "grouping up", as correctly chosen by DCU's system.

Sometimes, our system outperforms one of the online systems, but not both (examples 2 and 3). Google's translation of example 2 is stylistically sub-optimal due to the repetition, whereas Bing's translation of example 3 is neither adequate nor fluent.

Online systems outperform our system mainly in terms of fluency (examples 4, 5 and 6). In example 4, the online systems select a better preposition, in example 5 our system failed to generate

the imperative form and generated a gerund instead, and the output of our system for example 6 is overly literal.

**Table 10.** Translation examples for French-to-English.

| System | Example |
|---|---|
| (1) source | interdiction de **se regrouper** |
| (1) reference | ban on **gatherings** |
| google | ban on regrouping |
| bing | prohibition of regrouping |
| dcu (best) | prohibition to group up |
| (2) source | Utiliser un mouchoir à usage unique et le jeter |
| (2) reference | Use single-use tissues and throw them away |
| google (worst) | Use and dispose of a disposable tissue |
| bing | Use a single-use handkerchief and discard it |
| dcu | Use a single-use tissue and dispose of it |
| (3) source | Comment **s'attrape** le coronavirus? |
| (3) reference | How **does a person catch** the Coronavirus? |
| google | How do you get coronavirus? |
| bing (worst) | How does coronavirus get? |
| dcu | How do I get coronavirus ? |
| (4) source | Que faire face **aux** premiers signes? |
| (4) reference | What should someone do **at** its first signs? |
| google | What to do about the first signs? |
| bing | What to do about the first signs? |
| dcu (worst) | what to do with the first signs ? |
| (5) source | **Tousser ou éternuer** dans son coude ou dans un mouchoir |
| (5) reference | **Cough or sneeze** into your sleeve or a tissue |
| google | Cough or sneeze into your elbow or into a tissue |
| bing | Cough or sneeze in your elbow or in a handkerchief |
| dcu (worst) | Coughing or sneezing in your elbow or in a tissue |
| (6) source | C'est le médecin qui décide **de faire le test ou non**. |
| (6) reference | It is up to a physician **whether or not to perform the test**. |
| google | It's the doctor who decides whether or not to take the test. |
| bing | It is the doctor who decides whether or not to take the test. |
| dcu (literal) | It is the doctor who decides to take the test or not. |

### 4.4.3. Spanish to English

In Table 11 we present examples of Spanish sentences translated into English by the various MT systems. In the first sentence we observe that "gravedad" (which in this context should be translated as "serious" when referred to symptoms or illness) is incorrectly translated as "gravity" by Google and Bing, whereas our system proposes the imperfect yet more accurate translation "feel serious for any other symptoms".

In the second example, the generated translations for "entra en contacto" are either "come in contact" or "come into contact". However, this structure requires a prepositional phrase (i.e., "with" and a noun), which the systems do not produce. Note also that in Google's translation, there is a mismatch in the pronoun agreement (i.e., "hands" is plural and "it" is singular). Our system also produces an additional mistake as it translates the word "usado" as "used", whereas in this context (i.e., "wear gloves") the term "wear" would be more accurate.

The last two rows of Table 11 present translations where both our system and Google produce translations that are similar to the references. In the third sentence, our system produces the term "guidance" as a translation of "orientaciones" as in the reference, and Google's system generates a similar term, "guidelines". In contrast, Bing produces the word "directions" which can be more

ambiguous. In the last example, Bing's system generates the phrase "Please note that" which is not present in the source sentence.

**Table 11.** Translation examples for Spanish-to-English.

| System | Example |
|---|---|
| (1) source | Si tienes sensación de falta de aire o **sensación de gravedad** por cualquier otro síntoma llama al 112. |
| (1) reference | If you have difficulty breathing or you **feel that any other symptom is serious**, call 112. |
| google | If you have a feeling of shortness of breath or a feeling of gravity from any other symptom, call 112. |
| bing | If you have a feeling of shortness of breath or a feeling of gravity from any other symptoms call 112. |
| dcu (best) | if you feel short of breath or feel serious for any other symptoms, call 112. |
| (2) source | Lave las manos si **entra en contacto**, aunque haya usado guantes |
| (2) reference | Wash your hands **after any contact**, even if you have been wearing glove |
| google | Wash your hands if it comes in contact, even if you have worn gloves |
| bing | Wash your hands if you come into contact, even if you've worn gloves |
| dcu (worst) | wash your hands if you come in contact, even if you have used gloves |
| (3) source | Siga las **orientaciones** expuestas arriba. |
| (3) reference | Follow the **guidance** outlined above. |
| google | Follow the guidelines outlined above. |
| bing | Follow the directions above. |
| dcu (best) | follow the guidance presented above. |
| (4) source | Tenga en la habitación productos de higiene de manos. |
| (4) reference | Keep hand hygiene products in your room |
| google (best) | Keep hand hygiene products in the room. |
| bing | Please note that hand hygiene products are in the room. |
| dcu (best) | keep hand hygiene products in the room. |

### 4.4.4. Italian to English

Overall, based on the manual inspection of 100 sentences from the test set, our Italian-to-English MT system shows generally accurate and mostly fluent output with only a few exceptions, e.g., due to the style that is occasionally a bit dry. The overall meaning of the sentences is typically preserved and clearly understandable in the English output. In general, the output of our system compares favourably with the online systems used for comparison, and does particularly well as far as correct translation of the specialized terminology is concerned, even though the style of the online MT systems tends to be better overall.

The examples for Italian-English are shown in Table 12, which includes the Italian input sentence, the English human reference translation, and then the output in English provided by Google Translate, Bing and our final system.

In example 1, even though the style of our MT system's English output is somewhat cumbersome (e.g., with the repetition of "cells", although this is found in the other outputs, too), the clarity of the message is preserved, and the translation of all the technical terminology such as "epithelial cells" and "respiratory and gastrointestinal tracts" is correct; interestingly, the MT output of our system pluralizes the noun "tracts", which is singular in the other outputs as well as in the reference human translation, but this is barely relevant to accuracy and naturalness of style.

Similarly, the style of the MT output of our system is not particularly natural in example 2, where the Italian source has a marked cleft construction that fronts the main action verb but omits part of its subsequent elements, as is frequent in newspaper articles and press releases; this is why the English MT output of our system wrongly has a seemingly final clause that gives rise to an incomplete sentence, which is a calque of the Italian syntactic structure, even though the technical terminology is translated correctly (i.e., "strain" for "ceppo"), and the global meaning can still be grasped with a small amount

of effort. By comparison, the meaning of Bing's output is very obscure and potentially misleading, and the verb tense used in Google Translate's output is also problematic and potentially confusing.

In the sample of 100 sentences that were manually inspected for Italian-English, only very minor nuances of meaning were occasionally lost in the output of our system, as shown by example 3. Leaving aside the minor stylistic issue in the English output of the missing definite article before the noun "infection" (which is common with Bing's output, the rest being identical across the three MT systems), the translation with "severe acute respiratory syndrome" in the MT output (the full form of the infamous acronym SARS) for the Italian "sindrome respiratoria acuta grave" seems preferable, and more relevant, than the underspecified rendition given in the reference with "acute respiratory distress syndrome", which is in fact a slightly different condition.

**Table 12.** Translation examples for Italian-to-English.

| System | Example |
|---|---|
| (1) source | Le cellule bersaglio primarie sono **quelle epiteliali** del **tratto respiratorio e gastrointestinale**. |
| (1) reference | **Epithelial cells** in the **respiratory and gastrointestinal tract** are the primary target cells. |
| google | The primary target cells are the epithelial cells of the respiratory and gastrointestinal tract. |
| bing | The primary target cells are epithelial cells of the respiratory and gastrointestinal tract. |
| dcu | primary target cells are epithelial cells of the respiratory and gastrointestinal tracts. |
| (2) source | A indicare il nome un gruppo di esperti incaricati di studiare il nuovo **ceppo** di coronavirus. |
| (2) reference | The name was given by a group of experts specially appointed to study the novel coronavirus. |
| google | The name is indicated by a group of experts in charge of studying the new coronavirus strain. |
| bing | The name is a group of experts tasked with studying the new strain of coronavirus. |
| dcu | to indicate the name a group of experts in charge of studying the new strain of coronavirus. |
| (3) source | **Nei casi più gravi**, **l'infezione** può causare polmonite, **sindrome respiratoria acuta grave**, insufficienza renale e persino la morte. |
| (3) reference | **In more serious cases**, **the infection** can cause pneumonia, **acute respiratory distress syndrome**, kidney failure and even death. |
| google | In severe cases, the infection can cause pneumonia, severe acute respiratory syndrome, kidney failure and even death. |
| bing | In severe cases, infection can cause pneumonia, severe acute respiratory syndrome, kidney failure and even death. |
| dcu | in severe cases, infection can cause pneumonia, severe acute respiratory syndrome, kidney failure, and even death. |
| (4) source | Alcuni **Coronavirus** possono essere trasmessi da persona a persona, di solito dopo un contatto stretto con un paziente infetto, ad esempio **tra** familiari o in **ambiente sanitario**. |
| (4) reference | Some **Coronaviruses** can be transmitted from person to person, usually after close contact with an infected patient, for example, between family members or in a **healthcare centre**. |
| google | Some Coronaviruses can be transmitted from person to person, usually after close contact with an infected patient, for example between family members or in a healthcare setting. |
| bing | Some Coronaviruses can be transmitted from person to person, usually after close contact with an infected patient, for example among family members or in the healthcare environment. |
| dcu | some coronavirus can be transmitted from person to person, usually after close contact with an infected patient, for example family members or in a healthcare environment. |

**Table 12.** *Cont*.

| System | Example |
| --- | --- |
| (5) source | Il periodo di incubazione rappresenta il periodo di tempo che intercorre fra il **contagio** e lo sviluppo dei sintomi clinici. |
| (5) reference | The incubation period is the time between **infection** and the onset of clinical symptoms of disease. |
| google | The incubation period represents the period of time that passes between the infection and the development of clinical symptoms. |
| bing | The incubation period represents the period of time between contagion and the development of clinical symptoms. |
| dcu | the incubation period represents the period of time between the infection and the development of clinical symptoms. |
| (6) source | Qualora la madre sia **paucisintomatica** e si senta in grado di gestire autonomamente il **neonato**, madre e neonato possono essere gestiti insieme. |
| (6) reference | Should the mother be **asymptomatic** and feel able to manage her **newborn** independently, mother and newborn can be managed together. |
| google | If the mother is symptomatic and feels able to manage the infant autonomously, mother and infant can be managed together. |
| bing | If the mother is paucysy and feels able to manage the newborn independently, the mother and newborn can be managed together. |
| dcu | if the mother is paucisymptomatic and feels able to manage the newborn independently, mother and newborn can be managed together. |

In example 3, a seemingly minor nuance of meaning is lost in the MT output "in severe cases", as the beginning of the Italian input can be literally glossed with "in the more/most serious cases"; the two forms of the comparative and superlative adjective are formally indistinguishable in Italian, so it is unclear on what basis the reference human translation opts for the comparative form, as opposed to the superlative. However, even in such a case the semantic difference seems minor, and the overall message is clearly preserved in the MT output.

As far as example 4 is concerned, the MT outputs are very similar and correspond very closely to the meaning of the input, as well as to the human reference translation. Interestingly, our system's output misses the plural form of the first noun ("some coronavirus" instead of "Some Coronaviruses" as given by the other two MT systems, which is more precise), which gives rise to a slight inaccuracy, even though the overall meaning is still perfectly clear. Another interesting point is the preposition corresponding to the Italian "tra familiari", which is omitted by our system, but translated by Google as "between family members", compared to Bing's inclusion of "among". Overall, omitting the preposition does not alter the meaning, and these differences seem irrelevant to the clarity of the message. Finally, the translation of "ambiente sanitario" (a very vague, underspecified phrase, literally "healthcare environment") is interesting, with both our system and Bing's giving "environment", and Google's choosing "setting". Note that all three MT outputs seem better than the human reference "healthcare centre", which appears to be unnecessarily specific.

With regard to example 5, the three outputs are very similar and equally correct. The minor differences concern the equivalent of "contagio", which alternates between "infection" (our system and Google's) and the more literal "contagion" (Bing, which seems altogether equally valid, but omits the definite article, so the style suffers a bit). Interestingly, Google presents a more direct translation from the original with "time that passes", while our system and Bing's omit the relative clause, which does not add any meaning. All things considered, the three outputs are very similar, and on balance of equivalent quality.

Finally, example 6 shows an instance where the performance of our system is better than the other two systems, which is occasionally the case especially with regard to very specialized terminology concerning the COVID-19 disease. The crucial word for the meaning of the sentence in example 6 is "paucisintomatica" in Italian, which refers to a mother who has recently given birth; this is a highly technical and relatively rare term, which literally translates as "with/showing few symptoms".

Our system translates this correctly with the English equivalent "if the mother is paucisymptomatic", while Google gives the opposite meaning by missing the negative prefix, which would cause a serious misunderstanding, i.e., "If the mother is symptomatic", and Bing invents a word with "if the mother is paucysy", which is clearly incomprehensible. Interestingly, the human reference English translation for example 6 gives an overspecified (and potentially incorrect) rendition with "If the mother is asymptomatic", which is not quite an accurate translation of the Italian original, which refers to mothers showing few symptoms. The remaining translations in example 6 are broadly interchangeable, e.g., with regard to rendering "neonato" (literally "newborn") with "infant" (Google Translate) or "newborn" (our system and Bing's); the target sentences are correct and perfectly clear in all cases.

## 5. Making the Engines Available Online

We expose our MT services as webpages which can be visited by users on a worldwide basis. As there are 8 language pairs, in order to ensure provision of sufficiently responsive speed, we deploy our engines in a distributed manner. The high-level architecture of our online MT systems is shown in Figure 1. The system is composed of a webserver and separate GPU MT servers for each language pair. The webserver provides the user interface and distributes translation tasks. Users visit the online website and submit their translation requests to the webserver. When translation requests come in from a user, the webserver distributes the translation tasks to each MT server. The appropriate MT server carries out the translation and returns the translation result to the webserver, and the output is then displayed to the user.

Figure 2 shows the current interface to the 8 MT systems. The source and target languages can be selected from drop-down menus. Once the text is pasted into the source panel, and the ⟨Translate⟩ button clicked, the translation is instantaneously retrieved and appears in the target-language pane. We exemplify the system performance with a sentence—at the onset of the outbreak of the virus, a sentence dear to all our hearts—taken from *Die Welt* on 20th March 2020.

The translation process is portrayed in more detail in Figure 3. The DNT/TAG/URL module first replaces DNT ('do not translate') items and URLs with placeholders. It also takes care of tags that may appear in the text. The sentence-splitter splits longer sentences into smaller chunks. The tokeniser is an important module in the pipeline, as it is responsible for tokensing words, and separating tokens from punctuation symbols. The segmenter module segments words for specific languages. The compound splitter splits compound words for specific languages like German which fuse together constituent parts to form longer distinct words. The character normaliser is responsible for normalising characters. The lowercaser and truecaser module is responsible for lowercasing or truecasing words. The spellchecker checks for typographical errors and corrects them if necessary. Corresponding to these modules, we need a set of tools which perform the opposite operations: a detruecaser/recaser for detruecasing or recasing words after translation; a character denormaliser for certain languages; a compound rejoiner to produce compound words from subwords when translating into German; a desegmenter for producing a sequence from a set of segments; a detokeniser, which reattaches punctuation marks to words; and the DNT/TAG/URL module reinstates 'do not translate' entities, tags and URLS into the output translations.

At the time of writing, each engine had been queried, with the most preferred source language being English (92% of the time), and French the most in-demand target language (80% of the time). The most queried engine (English-to-French) had translated over 100,000 words. To date, therefore, the load on the systems has been relatively low, but we expect translation volume to increase as word gets around. Note that these statistics (language direction, and throughput in terms of words translated) are the only ones that we store, so we firmly believe our systems to be a realistic alternative to the freely available online systems that we use as a comparison in Section 4.2; access is free, there are no limits on throughput, and no personal data is retained by us for further processing.

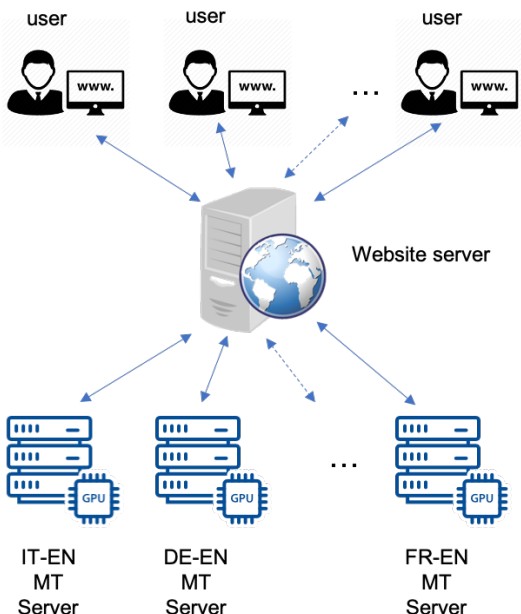

**Figure 1.** Architecture of the ADAPT COVID-19 Online MT System.

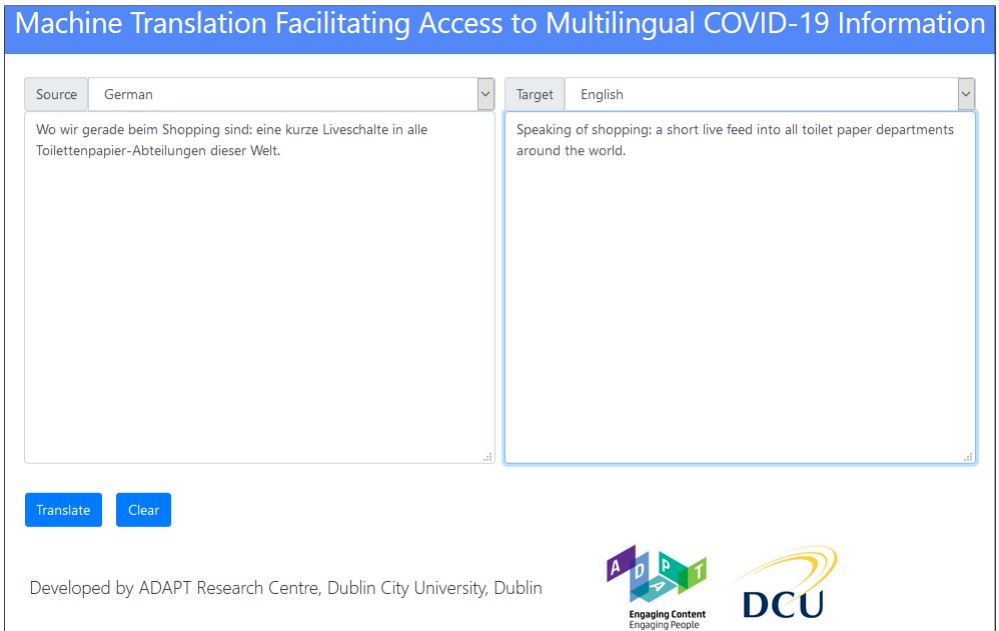

**Figure 2.** ADAPT COVID-19 Online MT System GUI, showing a sentence from *Die Welt* from 20/3/2020 translated into English.

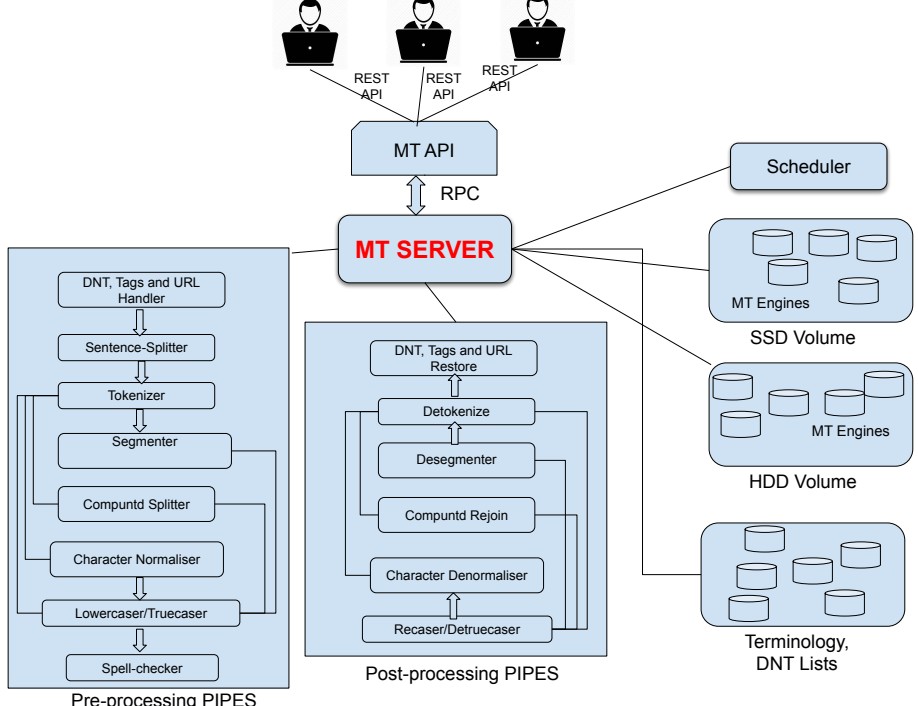

**Figure 3.** Skeleton of Online MT Service.

## 6. Concluding Remarks

This paper has described the efforts of the ADAPT Centre MT team at DCU to rapidly build a suite of 8 NMT systems for use both by medical practitioners and the general public to access multilingual information related to the COVID-19 outbreak in a timely manner. Using freely available data only, we quickly built MT systems for French, Italian, German and Spanish to/from English using a range of state-of-the-art techniques. In testing these systems, we demonstrated similar—and sometimes better—performance compared to popular longstanding online MT engines.

In a complementary human evaluation, we demonstrated the strengths and weaknesses of our engines compared to Google Translate and Bing Translator for a range of translational phenomena.

Finally, we described how these systems can be accessed safely online, with the intention that people can quickly access important multilingual information that otherwise might have remained hidden to them because of language barriers in a secure and confidential manner.

As in this paper, our future work has two main areas of attention. From an engineering perspective, we intend to explore whether multilingual models could beat our individual MT systems for specific language pairs, using transfer learning techniques to fine-tune on the different language pairs. We demonstrated clearly that adding all available data without due consideration caused translation performance to deteriorate, but it may be the case that adding the full EMEA set could improve performance. From a human evaluation point of view, given the automatic evaluation scores obtained, it is clear that the output by DeepL deserves further inspection. More generally, of course, more language pairs can be added, and larger amounts of in-domain training data are now publicly available which may lead to further performance gains.

**Author Contributions:** Conceptualization, A.W.; Formal analysis, A.W., M.P. and A.P.; Methodology, R.H., G.X., M.P. and A.P.; Resources, R.H. and M.P.; Software, G.X.; Validation, R.H. and F.G.; Writing–original draft, A.W., R.H. and F.G.; Writing–review & editing, A.W. and R.H. All authors have read and agreed to the published version of the manuscript.

**Funding:** The ADAPT Centre for Digital Content Technology is funded under the Science Foundation Ireland (SFI) Research Centres Programme (Grant No. 13/RC/2106) and is co-funded under the European Regional Development Fund.

**Acknowledgments:** Many thanks to the five anonymous reviewers for their helpful suggestions as to how to improve this paper. Any remaining errors are of course our own fault.

**Conflicts of Interest:** The authors declare no conflict of interest.

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
