# Peer review of "Rapid Development of Competitive Translation Engines for Access to Multilingual COVID-19 Information"

_informatics, doi:10.3390/informatics7020019_

Round 1
Reviewer 1 Report
The paper present the development of a machine translation system and an adaptation to a particular domain, specifically texts regarding the disease COVID-19.
Overall, I find the paper of good quality, but I would recommend minor revisions before publication, as I have a few specific comments regarding the content of the article.
1.) On page 1, lines 29 and 30: the mentioned countries in this sentence are Spain, France, and Italy, but the mentioned languages are Spanish, French, and German. I suspect this to be an error.
2.) On page 3, lines 101 and 102: it is mentioned that NMT typically requires larger amounts of training data. Can you provide some reference to an estimate of the size training data where NMT starts outperforming SMT?
3.) Table 4: the table differs from other tables (specifically 1, 2, 3, and 7). In other tables, the language pairs are in the table rows, while in table 4, they are in a 2x2 manner. This makes it more difficult to compare the data between tables. Also, data in other tables are always given with two decimal places, but not here.
4.) Page 6, line 177: given the numbers in the table, should the number here not be 1.9 instead od 2.1.
5.) Page 6, the paragraph from line 184 to 186: I am not sure if this holds. Please double-check the data in the tables.
6.) Page 8: there are some mentions of statistical significance, but there is no mention of which test was used and what the p-values are.
7.) Figure 3: The image is not well decipherable. I would suggest enlarging the figure to page width.
8.) The results largely concern translation from other languages to English. I believe the authors should address why there are no results regarding the translation from English is not evaluated as much (yet). Also, examples from Italian is much more described than examples from other languages.
On a remark not directly related to the publishment of this paper: the system is designed so that each language works on a separate server. If there were large differences in the translation demand for different language pairs, it would mean that the servers are unequally loaded. I assume the demand can not be determined at this time. However, after a sufficient time, it might be interesting to see data which language pairs are more in demand for translation of COVID related texts.
Author Response
1.) On page 1, lines 29 and 30: the mentioned countries in this sentence are Spain, France, and Italy, but the mentioned languages are Spanish, French, and German. I suspect this to be an error.
Response: Thanks, that was an error. Corrected now!
2.) On page 3, lines 101 and 102: it is mentioned that NMT typically requires larger amounts of training data. Can you provide some reference to an estimate of the size training data where NMT starts outperforming SMT?
Response: inserted some NMT vs SMT comparisons from Koehn and Knowles (2017)
3.) Table 4: the table differs from other tables (specifically 1, 2, 3, and 7). In other tables, the language pairs are in the table rows, while in table 4, they are in a 2x2 manner. This makes it more difficult to compare the data between tables. Also, data in other tables are always given with two decimal places, but not here.
Response: Fixed these
4.) Page 6, line 177: given the numbers in the table, should the number here not be 1.9 instead od 2.1.
Response: Thanks, that was an error. Corrected now!
5.) Page 6, the paragraph from line 184 to 186: I am not sure if this holds. Please double-check the data in the tables.
Response: checked, and we believe what's in the text is correct
6.) Page 8: there are some mentions of statistical significance, but there is no mention of which test was used and what the p-values are.
Response: we have added both of these in the new version
7.) Figure 3: The image is not well decipherable. I would suggest enlarging the figure to page width.
Response: made larger as suggested
8.) The results largely concern translation from other languages to English. I believe the authors should address why there are no results regarding the translation from English is not evaluated as much (yet). Also, examples from Italian is much more described than examples from other languages.
Response: explained in the new version of the paper
On a remark not directly related to the publishment of this paper: the system is designed so that each language works on a separate server. If there were large differences in the translation demand for different language pairs, it would mean that the servers are unequally loaded. I assume the demand can not be determined at this time. However, after a sufficient time, it might be interesting to see data which language pairs are more in demand for translation of COVID related texts.
Response: we have added in some usage statistics of the systems so far
Reviewer 2 Report
In the paper “Facilitating access to Multilingual Covid-19 Information via Neural Machine Translation” the authors build a range of NMT systems to facilitate access to multilingual content related to COVID-19.
The authors chose to build 8 NMT systems: DE,FR,IT,ES ↔ EN. The engines are made publicly available, so users can access multilingual information related to COVID-19.
The authors train MT systems using relatively large parallel corpora, comprising of several million generic segments, to which some domain specific data is added (TAUS Corona dataset, EMEA corpus).
Next, a well known sentence selection technique (Axelrod et al) is used to select pseudo in-domain segments from generic parallel datasets (ParaCrawl, Wikipedia).
Evaluated on a domain specific test set, the authors show that the performance of the resulting NMT system is comparable to online MT platforms.
The presented research is in general well conducted, clear and the paper is well written. The only concern, however, is that the presented work is not particularly new. The presented techniques are well known and widely used among “MT practitioners”.
On the other hand, it is appreciated that the authors present the used techniques in a structured and clear fashion, and use a variety of methods to tackle a very relevant use case.
Apart from these general observations, I have the following remarks:
-Table 1. Only 3 datasets are mentioned (TAUS corona, EMEA and Sketch), which all contain in-domain data. The other generic datasets used should also be listed. Now it is unclear what other data (open-source , in house data?) is used for training the baseline engines, making it impossible to reproduce the results presented in the paper.
It is also worrying that “only” 10M parallel segments are used for training ES,FR,IT↔EN baseline engines, because more high quality open source parallel data is available for these language pairs, which could boost the performance of the system, make it suffer less from overfitting on the in-domain dataset, and make the translations more fluent.
-Table 2. The authors test the NMT system on a held out test set (TAUS) and a separate smaller test set (‘Reco’). The ‘Reco’ test set is described as a ‘separate test set specifically on COVID-19 recommendations.’ It is not clear what is meant by this. The authors should describe this test set in more detail. Are these recommendations scraped from the web? What is the criterion to add a segment to this test set? The size of this test set is very small (only 73 sentences ES-EN?), especially because it is the only test set used for comparison to online MT-platforms.
-Table 4. It is unclear why the online MT-platforms are only evaluated on the small ‘Reco’ test sets. Is there a reason why the performance is not compared against the TAUS test set? It could also be interesting to compare against a generic, out of domain test set.
While the comparison to ‘leading online MT systems’ is appreciated, the authors do not compare the performance of their engines against, possibly, the single leading MT system: DeepL. Is there a reason for this?
Line 205. From Table 1 it seemed that the EMEA corpus was already added to the baseline training data. However, only part of the EMEA corpus is used (via the well known sentence selection approach Axelrod et al).
Line 217. What do the authors mean with “Fine-tuned on the training data from the Taus Corona crisis corpus along with 300k sentence-pairs from the EMEA corpus..”. Does this mean that the model is trained for some extra iterations, only on the TAUS+EMEA(300k) corpus? Or does it mean that it is trained for some extra iterations on a well considered sample of in-domain + generic data? The latter strategy seems recommended, to avoid overfitting on the TAUS+EMEA corpus. The authors could be more clear. Same remark for sentence at line 241.
Table 6:
-Engine 2 (1+EMEA). Possibly, adding all EMEA (around 1M segments) data to the baseline data would result in similar, or even better performance. The system would probably also benefit from the +/-700k sentences from the medical domain now thrown away for no good reason.
-Engine 3 (1+SEC). In Khayrallah and Koehn (2018) it has been shown that untranslated sentences (i.e. copying target sentences to the source) can severely negatively impact NMT systems in mid to high resource scenarios. The recommended strategy would have been to backtranslate the English monolingual data to the source language (using Engine 2, or Engine 5), or to not use the monolingual data at all. I see no reason to believe that the approach of Currey et al. can be viewed as a worthwhile alternative to backtranslation (in mid to high resource scenario’s).
As a final general remark I was wondering why the authors did not chose to build multilingual systems, as in Arivazhagan et al (2019)? As these domain specific MT systems needed to be build and deployed as rapid as possible, a multilingual approach seems a logical choice. In a second step, transfer learning techniques could have been used to fine tune on the different language pairs (as in Hu et al 2018).
Using multilingual systems and transfer learning techniques, it would be easier to support a wider range of languages. Via zero-shot translation, it would then also be possible to support translation to and from other languages than English, being very relevant for this use case.
Author Response
-Table 1. Only 3 datasets are mentioned (TAUS corona, EMEA and Sketch), which all contain in-domain data. The other generic datasets used should also be listed. Now it is unclear what other data (open-source , in house data?) is used for training the baseline engines, making it impossible to reproduce the results presented in the paper.
Response: in the original paper, we described in lines 207-214 what generic data we used. Note too that in line 235, we stated that we took five million low-scoring sentence-pairs. However, we have clarified this further in the final version.
As to the replicability of our results, the reviewer is correct. If we are able to release the data sets we actually used, we will, although there may be copyright and other licensing issues involved here ...
It is also worrying that “only” 10M parallel segments are used for training ES,FR,IT↔EN baseline engines, because more high quality open source parallel data is available for these language pairs, which could boost the performance of the system, make it suffer less from overfitting on the in-domain dataset, and make the translations more fluent.
Response: we do not believe that 10M sentence-pairs is small (as implied by the insertion of “only”). The suggestion to avoid overfitting is a good one, and we have incorporated this comment into the paper. Many thanks!
-Table 2. The authors test the NMT system on a held out test set (TAUS) and a separate smaller test set (‘Reco’). The ‘Reco’ test set is described as a ‘separate test set specifically on COVID-19 recommendations.’ It is not clear what is meant by this. The authors should describe this test set in more detail. Are these recommendations scraped from the web? What is the criterion to add a segment to this test set? The size of this test set is very small (only 73 sentences ES-EN?), especially because it is the only test set used for comparison to online MT-platforms.
Response: we have described the Reco test sets and their provenance in the revised version of the paper.
-Table 4. It is unclear why the online MT-platforms are only evaluated on the small ‘Reco’ test sets. Is there a reason why the performance is not compared against the TAUS test set? It could also be interesting to compare against a generic, out of domain test set.
Response: We explain why it is important to use these test sets in the revised version of the paper.
While the comparison to ‘leading online MT systems’ is appreciated, the authors do not compare the performance of their engines against, possibly, the single leading MT system: DeepL. Is there a reason for this?
Response: given the very quick turnaround requirement by the journal, we haven't been able to perform an in-depth comparison against DeepL. However, we have obtained automatic evaluation scores, and you are right: they are higher. Unfortunately, the online version of the system is almost unusable, however ...
Line 205. From Table 1 it seemed that the EMEA corpus was already added to the baseline training data. However, only part of the EMEA corpus is used (via the well known sentence selection approach Axelrod et al).
Response: as stated earlier, we have tried to clarify what data was used to train the systems.
Line 217. What do the authors mean with “Fine-tuned on the training data from the Taus Corona crisis corpus along with 300k sentence-pairs from the EMEA corpus..”. Does this mean that the model is trained for some extra iterations, only on the TAUS+EMEA(300k) corpus? Or does it mean that it is trained for some extra iterations on a well considered sample of in-domain + generic data? The latter strategy seems recommended, to avoid overfitting on the TAUS+EMEA corpus. The authors could be more clear. Same remark for sentence at line 241.
Response: your interpretation is correct,and we have clarified this in the paper. Thanks for the suggestion.
Table 6:
-Engine 2 (1+EMEA). Possibly, adding all EMEA (around 1M segments) data to the baseline data would result in similar, or even better performance. The system would probably also benefit from the +/-700k sentences from the medical domain now thrown away for no good reason.
Response: we have added the following: "It is an open question as to whether including the full EMEA set would improve performance, but adding in the 300K most similar sentences barely helps; even if we assume that 700K sentence-pairs are disregarded by our approach, after cleaning and other filtering, the remaining subset would be far smaller than 700K. Nonetheless, we leave this experiment for future work, just in case the additional data does prove to be effective."
-Engine 3 (1+SEC). In Khayrallah and Koehn (2018) it has been shown that untranslated sentences (i.e. copying target sentences to the source) can severely negatively impact NMT systems in mid to high resource scenarios. The recommended strategy would have been to backtranslate the English monolingual data to the source language (using Engine 2, or Engine 5), or to not use the monolingual data at all. I see no reason to believe that the approach of Currey et al. can be viewed as a worthwhile alternative to backtranslation (in mid to high resource scenario’s).
Response: we cite Khayrallah and Koehn (2018), and show that their arguments tally with what we did in the paper. Thanks for the pointer.
As a final general remark I was wondering why the authors did not chose to build multilingual systems, as in Arivazhagan et al (2019)? As these domain specific MT systems needed to be build and deployed as rapid as possible, a multilingual approach seems a logical choice. In a second step, transfer learning techniques could have been used to fine tune on the different language pairs (as in Hu et al 2018).
Response: we cite Arivazhagan et al (2019), and give two other citations which show that massively multilingual systems are not guaranteed to beat language-pair-specific systems. There are also, of course, hardware concerns when trying to build massively multilingual systems, and we raise these in this context in the revised version of the paper. Finally, we do add transfer learning as a technique to pursue in future work. Thanks for the suggestion.
Reviewer 3 Report
It is nice to see that by using cutting-edge NMT techniques to solve real problems, and improve the performance by a pretty sizable margin compared with the baseline. There is no particular innovation at the model level, however, the author mitigated the problem of domain inconsistency between general and COVID-19 through data selection. As a result, the quality of translation on the COVID-19 domain has improved somewhat through data selection, but it is still lower than the mainstream translation engines on the market, which makes the value of this work somewhat questionable. This is also true in the manual evaluation section. If most people just think that this work is equivalent to Google and other translation engines, what is the research value of this work? The author's project attempt deserves encouragement, but as an academic paper, there are still some places worth discussing. I hope that the author can explore a variety of domain adaptation strategies, which should go beyond the translation engine of the general domain, or explore a larger scope, such as document translation and so on.
Question:
Table 5. Is the data crawled in Words or Sentences? If it's Words, English has too little data.
Author Response
As a result, the quality of translation on the COVID-19 domain has improved somewhat through data selection, but it is still lower than the mainstream translation engines on the market, which makes the value of this work somewhat questionable. This is also true in the manual evaluation section. If most people just think that this work is equivalent to Google and other translation engines, what is the research value of this work? The author's project attempt deserves encouragement, but as an academic paper, there are still some places worth discussing.
Response: there are of course many reasons why people do not like using freely available engines such as GT (e.g. security). We never set out to try to replicate what massive corporations are able to do; we cannot compete with them at that level. But we do not track who uses our systems, we do not retain any data, and only measure the per-engine throughput for our own purposes. If nothing else, our comparisons show how good these engines actually are, but also that using clever techniques -- albeit well-understood ones -- much smaller academic groups can get close to -- and in some cases beat -- the performance of freely available engines.
I hope that the author can explore a variety of domain adaptation strategies, which should go beyond the translation engine of the general domain, or explore a larger scope, such as document translation and so on.
Response: the journal gave us 5 days to turn the paper around, so unfortunately no new experiments were possible in that time.
Table 5. Is the data crawled in Words or Sentences? If it's Words, English has too little data.
Response: at the time, this figure was correct. It may partly explain why COVID-19 has been such a problem in the UK and the US, as at the time, there really was very little data related to COVID-19 available on the NHS website.
Reviewer 4 Report
- Abstract: Please make clear what text types can be covered by your MT system (news paper articles [politics, restrictions, laws, etc.], popular science articles, medical information, scientific articles, all of those or even more?) - you mention the text types in lines 35-36, but I think the information would be valuable in the abstract
- Introduction: Maybe reconsider if you really want to motivate your language selection via which countries coped well/badly in the Corona crisis. I think it would be much less judgemental if you wrote somethink like "the US, the UK, France, Spain, Italy and Germany are very effected by the Corona crisis, therefore we want to include English, French, German, Spanish, and Italian in our MT system to enable knowledge exchange" (at the moment - 20th May - the US, Russia and Brazil are most affected and it all started in China, so maybe you can find a better line of argumentation)
- line 21-26: maybe rethink the tense, with publication the events have past (even with fast publication)
-
line 55-66: consider paraphrasing and summarising the quotes
-
line 62 & 65: please use same refernce style ((Disaster 2.0, 2011) vs. Brendan McDonald, UN OCHA in Disaster 2.0 (2011))
-
line 75-87: Do you really need the numbering system from the paper? For me, it's quite difficult to ready as the citations are also presented in these brackets; or maybe write "Recommendation [7a] notes that “Translating..." instead of "[7a] notes that “Translating..."
Section 4.4:
- Could you provide some brief information in the introduction why you chose the examples you are presenting in the following?
- the explainations on Italian-English examples are presented with much more detaila than is done for the other languages. Could you adjust this or explain, why you focus on Italian?
- table 10 misses some colour-coding (ex. 2,3)
table 11: misses almost all colour-coding (uses only bold)
table 12: misses all colour-coding
-> please adjust and use the same coding in all tables
in general:
- provide page numbers for direct quotes, you did so for some quotes, but not for all
- Concerning language in style: I suggested some improvements and I think some semntence structures could be simplified or seem a bit clumbsy (esp. in the first sections), but I'm not a native speaker. So please ignore any suggestions you don't agree with

Author Response
Abstract: Please make clear what text types can be covered by your MT system (news paper articles [politics, restrictions, laws, etc.], popular science articles, medical information, scientific articles, all of those or even more?) - you mention the text types in lines 35-36, but I think the information would be valuable in the abstract
Response: we have added those to the Abstract, too.
Introduction: Maybe reconsider if you really want to motivate your language selection via which countries coped well/badly in the Corona crisis. I think it would be much less judgemental if you wrote somethink like "the US, the UK, France, Spain, Italy and Germany are very effected by the Corona crisis, therefore we want to include English, French, German, Spanish, and Italian in our MT system to enable knowledge exchange" (at the moment - 20th May - the US, Russia and Brazil are most affected and it all started in China, so maybe you can find a better line of argumentation)
Response: we believe our revised version is less judgemental. Thanks for the suggestion; we're not trying to score points, here!
line 21-26: maybe rethink the tense, with publication the events have past (even with fast publication)
Response: fixed!
line 55-66: consider paraphrasing and summarising the quotes
Response:we have kept the quotes, but have added page numbers throughout.
line 62 & 65: please use same refernce style ((Disaster 2.0, 2011) vs. Brendan McDonald, UN OCHA in Disaster 2.0 (2011))
Response: the references were actually part of the quotes, but that's fixed now.
line 75-87: Do you really need the numbering system from the paper? For me, it's quite difficult to ready as the citations are also presented in these brackets; or maybe write "Recommendation [7a] notes that “Translating..." instead of "[7a] notes that “Translating..."
Response: we have added "Recommendation" in each case, and added the appropriate page number
Section 4.4:
Could you provide some brief information in the introduction why you chose the examples you are presenting in the following?
Response: we have fixed this in the revised version of the paper.
the explainations on Italian-English examples are presented with much more detaila than is done for the other languages. Could you adjust this or explain, why you focus on Italian?
Response: explained in the revised version of the paper.
table 10 misses some colour-coding (ex. 2,3)
table 11: misses almost all colour-coding (uses only bold)
table 12: misses all colour-coding
-> please adjust and use the same coding in all tables
Response: harmonised for each table now.
in general:
provide page numbers for direct quotes, you did so for some quotes, but not for all
Response:all direct quotes now have page numbers. Thanks for pointing this out!
Reviewer 5 Report
Here are some comments for the manuscript:
- Table1
More details need to be added to table1, i.e DE-EN, write full term since that you did not mention in text.
- Table 3,4
Hard to follow and compare two tables. Maybe it will be better to combine two table.
- Section 4.2
I like this section, however, since your method was almost the worst, probably, you can make it short. Many details had been added. You can make it better. Or you can combine it with section 4.1 as one section.
- Table6
The table showed just one translation It-EN. But you did not state why or at least mention that in statement you select this as an example.
- Table2 6-11
The DCU term is first time appears. You need to declared in text as your and as your system.
Author Response
- Table1
More details need to be added to table1, i.e DE-EN, write full term since that you did not mention in text.
Response: we use the internationally recognised ISO 639-1 language codes, which we believe are well understood.
- Table 3,4
Hard to follow and compare two tables. Maybe it will be better to combine two table.
Response: the first table relates to data sizes, and the second table to MT evaluation scores, so we have left them as in the original paper.
- Section 4.2
I like this section, however, since your method was almost the worst, probably, you can make it short. Many details had been added. You can make it better. Or you can combine it with section 4.1 as one section.
Response: thanks for your comment. It's only half a page, so we have left the section as is.
- Table6
The table showed just one translation It-EN. But you did not state why or at least mention that in statement you select this as an example.
Response: many of the reviewers have pointed this out. We have motivated this better in the revised version of the paper.
- Table2 6-11
The DCU term is first time appears. You need to declared in text as your and as your system.
Response: We have fixed this in the revised version of the paper. Thanks for the observation.
Round 2
Reviewer 3 Report
The author does not seem to have answered the last question. In addition, "... demonstrates clearly how strong the online baseline engines are on this test set. However, in the next section, we run a set of experiments which showed Improved performance of our systems, In some cases comparable to those of these online engines "(line 215-217) show that the model is only comparable to mainstream commercial engines, and no any new is proposed. The question is, why should we choose a translation engine proposed by this paper instead of a free commercial translation engine with more guaranteed quality? The comparison of other translation examples cannot explain the problem. A new metric is needed to support these examples, such as low-frequency word statistics. In the absence of such metrics, there is no persuasive argument.
Author Response
Reviewer 3: The question is, why should we choose a translation engine proposed by this paper instead of a free commercial translation engine with more guaranteed quality?
Response: in the last version, at the end of the penultimate paragraph in the Introduction, we had added the sentence:
"we are allowing users to avail of high-quality MT with none of the usual privacy concerns associated with using online MT platforms".
However, we acknowledge that this was hard to see, and may not have been a complete answer to the reviewer's question.
Privacy is just one reason why people prefer not to access tools such as Google Translate; all data processed by the engine is retained by Google (and the same is true for the other multinational companies) for its own use, so there is huge data leakage from people using such systems without having read the small print. Also, many people prefer for their interactions with Google tools to not be used for the direction of more (supposedly personalised) ads to be pointed in their direction.
We could flesh this out if you prefer, but we have chosen not to be this blunt in this revised version. Instead, we have added the following sentence at the end of Section 5:
"Note that these statistics (language direction, and throughput in terms of words translated) are the only ones that we store, so we firmly believe our systems to be a realistic alternative to the freely available online systems that we use as a comparison in Section~\ref{stackup}; access is free, there are no limits on throughput, and no personal data is retained by us for further processing."
We also added the word "safely" and the phrase "in a secure and confidential manner" to the following sentence in the Conclusion:
"Finally, we described how these systems can be accessed safely online, with the intention that people can quickly access important multilingual information that otherwise might have remained hidden to them because of language barriers in a secure and confidential manner."
Finally, as another benefit of building systems oneself is that the systems can learn and be retrained based on feedback, at the end of the Introduction, we have added the following sentence:
"Finally, of course, by building our own engines, we retain control over the systems, which facilitates continuous incremental improvement of the models via feedback and by the availability of additional training data, or novel scientific breakthroughs in the field of Neural MT (NMT)."